# Gut Microbiome–Estrobolome Profile in Reproductive-Age Women with Endometriosis

**DOI:** 10.3390/ijms242216301

**Published:** 2023-11-14

**Authors:** Angel Hsin-Yu Pai, Yi-Wen Wang, Pei-Chen Lu, Hsien-Ming Wu, Jia-Ling Xu, Hong-Yuan Huang

**Affiliations:** 1Department of Obstetrics and Gynecology, Chang Gung Memorial Hospital, Linkou Medical Center, Taoyuan City 333423, Taiwan; 2College of Medicine, Chang Gung University, Taoyuan City 33302, Taiwan

**Keywords:** gut microbiota, endometriosis, dysbiosis, β-glucuronidase, 16S ribosomal-RNA gene, estrogen metabolites

## Abstract

Microbiota is associated with our bodily functions and microenvironment. A healthy, balanced gut microbiome not only helps maintain mucosal integrity, prevents translocation of bacterial content, and contributes to immune status, but also associates with estrogen metabolism. Gut dysbiosis and estrobolome dysfunction have hence been linked to certain estrogen-dependent diseases, including endometriosis. While prior studies on microbiomes and endometriosis have shown conflicting results, most of the observed microbial differences are seen in the genital tract. This case-control study of reproductive-age women utilizes their fecal and urine samples for enzymatic, microbial, and metabolic studies to explore if patients with endometriosis have distinguishable gut microbiota or altered estrogen metabolism. While gut β-glucuronidase activities, microbial diversity, and abundance did not vary significantly between patients with or without endometriosis, fecal samples of patients with endometriosis were more enriched by the *Erysipelotrichia* class and had higher folds of four estrogen/estrogen metabolites. Further studies are needed to elucidate what these results imply and whether there indeed is an association or causation between gut microbiota and endometriosis.

## 1. Introduction

Human microbiota, which comprises bacteria, archaea, protists, fungi, and viruses that reside both in and on our bodies, is vital for bodily functions and wellbeing [1]. With the gut harboring most of the symbiotic microbial cells, its development co-evolves with numerous host determinants and is unique for each individual [2]. Not only do gut microbial communities digest food and make essential vitamins, they also play important roles in maintaining homeostasis, immune function, metabolism, and many other functions [3]. Emerging evidences suggest that dysbiosis, which is often observed with reductions in overall microbial diversity or alterations in relative bacterial abundances in the gut [4,5], contributes to many pathological conditions [4,6,7,8,9,10,11], such as inflammatory bowel disease, metabolic syndromes, autoimmune disorders, cancers, and, of particular interest, endometriosis [12,13,14,15].

Endometriosis, a common yet perplexing gynecological disease, causes a broad spectrum of symptoms, from pain to infertility, in affected women [16]. Globally, it affects approximately 10% of females of childbearing age [16,17], with the incidence rate reaching as high as 40~60% in those with abdominal pain and/or unexplained infertility [15]. Defined as the presence of endometriotic lesions outside of the uterus [12], it is an estrogen-dependent, multifactorial, and chronic condition influenced by genetic, immunologic, and environmental factors [12,15,18]. These extrauterine implants manifest in various forms [17], complicating its diagnosis and establishment of a unified classification system. Based on the number of lesions and depth of infiltration, the American Society for Reproductive Medicine (ASRM) has classified endometriosis into four stages with a scoring system: minimal (Stage I), mild (Stage II), moderate (Stage III), and severe (Stage IV). Many theories have been proposed regarding its pathogenesis, with the most widely accepted being “retrograde menstruation” [1]; the mechanisms of its development and progression, noninvasive diagnostic methods, and novel pharmacological treatments are still large areas of interest and research. 

A healthy, balanced state of gut biomass sustains the integrity of mucosal lining, protects against pathogenic insults, and regulates physiologic processes [14]. Dysbiosis, therefore, may impair epithelial integrity, resulting in translocation of bacteria or leakage of endotoxins [19]. Inciting a pro-inflammatory state and changing metabolite composition are associated with the induction or exacerbation of the disease [20]. Moreover, many intestinal bacteria have been found to carry estrogen-metabolizing enzymes [21,22], which influence the estrobolome (total collection of genes, in the gut microbiota, responsible for estrogen metabolism [1]) and, ultimately, estrogen-mediated disorders. For instance, bacterial species with β-glucuronidase can deconjugate the inactive, conjugated estrogen into the active form, thus increasing intestinal reabsorption and circulation of estrogen [23,24,25].

Studies on mice have demonstrated the development of a distinct composition of gut microbiota after 42 days of endometriosis induction [26] and possible dysbacteriosis due to altered fecal metabolomics [27]. In addition, another study further showed reduction in endometriotic lesion growth, proliferation, and inflammation after treatment with broad-spectrum antibiotics [28]. Yet, such changes in bacterial composition cannot be seen in early phases of lesion formation [26,29]. Studies on humans, on the other hand, have yielded more pronounced findings in endometriotic lesions [30] or the female reproductive tract [31] while results from gut microbiota have shown more inconsistencies [14,31,32]. Therefore, by focusing on a specific population of women of the same ethnicity, we aim to explore whether there is an association between gut microbiota and endometriosis, in terms of enzymatic expressions, bacterial compositions, and estrogen metabolite variations. 

## 2. Results

A total of 74 women (35 control, 37 endometriosis) were recruited based on the inclusion criteria. Due to inability to provide adequate urine or fecal samples, only data from 51 patients (24 control, 27 endometriosis) were used for analyses. All of the women were of Taiwanese ethnicity, and the baseline characteristics, detailed in Table 1, between the control and endometriosis group did not show significant differences in age (average 37.7 ± 1.3 years vs. 38.1 ± 1.0 years, *p* = 0.77) and body mass index (BMI, average 24.04 ± 0.87 kg/m^2^ vs. 21.77 ± 0.73, *p* = 0.051). The only significant difference between the two groups was the pre-operative CA-125 level, which, as expected, was much higher in the endometriosis group (21.54 ± 3.12 vs. 70.07 ± 9.07, *p* < 0.001). 

### 2.1. Similar ß-Glucuronidase Activity between Endometriosis and Control

Approximately 60 bacterial genera residing in the human intestinal tract encode β-glucuronidase and β-glucosidase [21]. While β-glucosidase, which serves as the control enzyme, catalyzes the last step in cellulose hydrolysis and is important for digestion of nutrients [22], β-glucuronidase can deconjugate the biologically inactive conjugated estrogen into the active, deconjugated form [24]. Enzymatic activity assays of fecal samples from the control and endometriosis group revealed insignificant differences between the average level of β-glucuronidase activity (1823.45 U/L vs. 1480.09 U/L, *p* = 0.35) and β-glucosidase (46.45 U/L vs. 33.92 U/L, *p* = 0.15) (Figure 1).

### 2.2. Patients with and without Endometriosis Shared Similar Bacterial Abundance, Diversity, Richness, and Evenness in the Gut

Compositions of microbial species at each hierarchical level was demonstrated with heat trees (Figure 2A,B) and bar charts (Appendix A), which revealed similar relative bacterial abundances, both collectively and in each taxon, for both groups of patients. Alpha diversity, which estimates microbial diversity, richness, and evenness within samples, was compared between the two groups using four metrics. No significant differences were seen in Shannon–Wiener diversity index (Figure 3A), Simpson diversity index (Figure 3B), Chao1 richness estimator (Figure 3C), and Good’s coverage index (Figure 3D). Microbial composition between different samples was evaluated with beta diversity. Based on species annotation and operational taxonomic unit (OTU) abundance information, the unweighted UniFrac distance (Figure 4A), weighted UniFrac distance (Figure 4B), and quantitative analysis with principal coordinates analysis (PCoA) plots (Figure 4C,D) were calculated, and all of them revealed insignificant differences between the two groups.

### 2.3. Although Dysbiosis Was Not Demonstrated in the Endometriosis Group, a Higher Composition of Certain Bacteria Was Detected

Healthy, diverse gut microbiota helps maintain the gut epithelial barrier and homeostasis [20], which composes predominantly of Bacteroidetes and Firmicutes phyla. A lower ratio of Firmicutes/Bacteroidetes (F/B) in the gut has been correlated with healthiness while a higher ratio indicates dysbiosis [1]. As represented in the box plot (Figure 5A), although the average F/B ratio of the endometriosis group was slightly higher than that of the control (0.81 vs. 0.73, *p* = 0.4269), it did not reach a statistical significance. In addition, neither the control nor endometriosis group was significantly enriched with aerobic (Figure 5B) or anaerobic (Figure 5C) bacteria.

Non-parametric tests using analysis of similarities (ANOSIM) and multi-response permutation procedure (MRPP) were applied to determine whether the differences between the endometriosis and control groups were significantly greater than the differences within each group. Both ANOSIM (R-value = −0.23, *p* = 0.824) and MRPP (A-value = −0.0004907, *p*-value 0.511) had results in the negative range, which suggested that differences within each group were greater than those between the two groups. 

To identify distinct gut microbiota associated with endometriosis, linear discriminant analysis with effect size (LEfSe), which employed the non-parametric factorial Kruskal–Wallis (KW) sum-rank test, was performed (Figure 5D). Gut microbiota in the endometriosis group had statistically significant enrichment of taxa including bacteria in the *Erysipelotrichia* class (*p* = 0.0286), *Erysipelotrichales* order (*p* = 0.0286), *Erysipelotrichaceae* family (*p* = 0.0286), *Eisenbergiella* genus (*p* = 0.0474), and *Hungatella* genus (*p* = 0.0497). 

The compositional changes in gut microbiota between the two groups at different taxon levels were also calculated using Welch’s t-test. At the class level (Figure 5E), the endometriosis group showed higher abundance of *Erysipelotrichia* (*p* = 0.036). At the order level (Figure 5F), *Erysipelotrichales* (*p* = 0.036) and *Micrococcales* (*p* = 0.039) were more abundant in the endometriosis group. At the family level (Figure 5G), *Erysipelotrichaceae* (*p* = 0.036) and *Micrococcaceae* (*p* = 0.039) were of higher abundance in the endometriosis group while *Mariniflaceae* (*p* = 0.019) predominated in the control group. At the genus level (Figure 5H), four genera (*UBA1819, Eisenbergiella, Hungatella,* and *Erysipelatoclostridium*) were more involved in the endometriosis group.

### 2.4. Differences in Estrogen Metabolites Were Seen in the Fecal but Not Urine Samples from Patients with Endometriosis

Out of the 14 estrogens/estrogen metabolites tested, fecal samples from patients with endometriosis had significantly higher folds of estriol (*p* = 0.011), 16-epiestriol (*p* = 0.018), 16alpha-hydroxyestrone (*p* = 0.016), and 2-methoxyestradiol (*p* = 0.035). Meanwhile, estrogen metabolites levels in the urine did not differ significantly in patients with or without endometriosis, as shown in Table 2.

## 3. Discussion

This study provided comprehensive bacterial analyses, along with enzymatic assays and targeted metabolites quantification, in an effort to find associations of gut microbiome with endometriosis. Although no significant changes in microbial richness, diversity, β-glucuronidase activities, and urinary estrogen metabolites were detected in the endometriosis group, fecal samples from patients with endometriosis were more enriched by certain bacteria (*UBA1819, Eisenbergiella, Hungatella,* and *Erysipelatoclostridium*) and had higher folds of four estrogen metabolites (estriol, 16-epiestriol, 16alpha-hydroxyestrone, and 2-methoxyestradiol). In line with previous studies, we believed that relationships between gut microbiota and endometriosis exist. However, the difficulties lay in standardizing the study and control subjects in order to generate a more robust and replicable data, as our analyses revealed that there were more variations in among the individuals than between the two study groups.

Gut microbial β-glucuronidase had been demonstrated by various studies as a key regulator in host estrogen metabolism [15,22,24,25]. Encoded by the GUS gene, which is more commonly found in *Firmicutes* phylum [33], β-glucuronidase removes glucuronic acid from conjugated substrates that are present in many biliary excreted metabolites, steroid hormones, and xenobiotics, and promotes their reabsorption into the enterohepatic circulation [21]. A recent study [25] with both human and mouse models demonstrated that β-glucuronidase promoted the proliferation and migration of endometrial stromal cells, increased the number and volume of endometriotic lesions, and contributed to macrophage dysfunction. Similar to our study, they found no significant differences in microbial diversity or abundances but instead, a significantly higher composition of the *Desulfovibrionia* class in patients with endometriosis. While they were able to demonstrate that β-glucuronidase was present at higher levels in human and mouse endometriotic lesions, we did not find similar patterns in the fecal samples of patients with endometriosis. 

Of more than 10^4^ bacteria residing in the human gut, more than 95% could be assigned to four major phyla: Bacteroidetes, Firmicutes, Actinobacteria, and Proteobacteria [20]. Homeostasis relied on maintaining an adequate diversity and abundance, and the ratio of Firmicutes/Bacteroidetes (F/B ratio) has been shown to be an indicator of such balance [34]. In addition to prior studies associating higher F/B ratio with obesity [35], hypertension [36], and irritable bowel syndrome [37], elevation of this index was also seen in mice with endometriosis [26]. Although the results of this current study did not display evident dysbiosis, indicated as a reduction in bacterial diversity or elevation in F/B ratio, in patients with endometriosis, significantly higher proportion of *Erysipelotrichia* class, which belongs to the *Firmicutes* phylum, was discovered.

*Erysipelotrichia*, which composes of many common bacterial species in the gut, has been shown to associate with a high-fat diet with increased composition [38], whereas depletion resulted in higher fecal concentration of sugar acids and sugar alcohol [39]. However, further studies are needed to explore whether it implicates other clinical significances. Interestingly, the Endobiota study, which compared the vaginal, cervical, and gut microbiota between women with stage III or IV endometriosis and healthy controls, observed microbial differences at the genus level in all three microenvironments [31]. In particular, women with stage III or IV endometriosis had predominantly *Shigella* and *Escherichia* in their stool microbiome, which were discrepant from our results. While both of those species could become pathogenic [40,41], the four genera detected (*UBA1819, Eisenbergiella, Hungatella,* and *Erysipelatoclostridium*) in our study have no strong clinical correlations yet.

Lastly, since prior studies had suggested that the enterohepatic recirculation of estrogen contributed to its systemic levels [15,42,43,44], we applied liquid chromatography/tandem mass spectrometry for highly sensitive and reproducible detection of parent estrogens (estradiol and estrone) and 12 estrogen metabolites in all fecal and urinary samples. This could reflect whether presence or absence of the disease was associated with the amount of estrogen excretion in the stool or urine. With prior studies displaying inconsistent results, the current study, through identifying significantly higher folds of four targeted metabolites in the fecal samples of the endometriosis group, also had a different variation in the metabolites that were elevated. Of note, as estrogen metabolite concentrations had not been observed to be strongly correlated between urine and serum [44,45], we also did not detect any significant differences in estrogens and estrogen metabolites in the urinary samples of the two groups. 

Our study has several limitations that are often inherent in cross-sectional studies of gut microbiota in clinical settings. As mentioned, gut microbiota varies among individuals and is under influences of many factors throughout a person’s life, such as mode of delivery and feeding, environment, diet, immunity, lifestyle, and so on [11]. Therefore, it is difficult to standardize all these factors when recruiting patients; thus, understandably, many confounders are at play when profiling each person’s gut microbiota. By focusing on a specific ethnicity of women with similar age and excluding some important and controllable factors, such as those with strict dietary restrictions or habits, systemic diseases, prior or present diagnoses of malignancies, antibiotics or probiotics usage in the three months prior to recruitment, and so on, we attempted to narrow the variations among the participants. Although prior studies have shown possible associations between BMI and altered gut microbiota [46,47], we did not apply this parameter as one of our exclusion criteria. As only two of the recruited patients had BMI greater than 30 kg/m2 and further assessment revealed a sedentary lifestyle instead of glucose intolerance, endocrinological, or other metabolic disorders as the main contributor, they were still included in their respective study groups (one in control, one in endometriosis). However, given the current evidence, future studies should consider adding BMI to eliminate as many possible confounding factors as possible. Meanwhile, since Taiwanese women have diets more in line with the traditional Chinese diet, which is a mixture of vegetables, meat, fruits, and rice, than with Western diets [48], this could reduce some of the differences caused by various dietary patterns. 

Usage of hormone medications was not an exclusion factor in this study as all participants were those opting for surgical interventions for benign gynecological diseases due to failure of medical treatment or intolerable symptoms. Therefore, seven of the patients in the endometriosis group had received GnRH agonists or progestin within 3 months of their surgeries while only one patient in the control group had hormone treatment for contraception. 

All patients in the endometriosis group had histologically proven endometriosis and documented revised American Fertility Society (rAFS) scores. Approximately 55.6% (15/27) of the patients in the endometriosis group had stage III or IV of the disease, whereas all patients in the study group of the Endobiota Study were those of stage III or IV endometriosis. Such difference could be part of the reason why different species of dominating bacteria in stool microbiota were observed. In addition, while their control group was composed of women who were assumed to be free of the disease due to the absence of symptoms or imaging findings, ours were all surgically and pathologically confirmed. Clearly distinguishing the two groups would help avoid underestimation of the differences seen between the two groups. However, since we included women of all stages, locations, and presentations of endometriosis, variations within the group could obscure the final outcomes. As indicated by the results of the ANOSIM and MRPP analyses, differences were greater among individuals than between groups. 

Many theories have been proposed to explain this complex disease, such as retrograde menstruation, embryonic rest theory, coelomic metaplasia, lymphovascular metastasis, and endometrial stem/progenitor cells [30]. With retrograde menstruation postulation, it had been highlighted that only about 10% of patients experiencing this developed endometriosis [49]; therefore, other factors, including but not limited to anatomy, genetic, environment, lifestyle, menstrual cycle, and immunity were definitely at play. Justifiably, to identify the association between endometriosis and gut microbiota, subsequent studies would benefit if more stringent patient selection and study design were applied. With larger, well-controlled studies, the association or causation, if so, which direction of the causation, between endometriosis and gut microbiota could be better elucidated and such information would help provide novel insights into diagnostic methods or therapeutics for this complex disease.

## 4. Materials and Methods

### 4.1. Study Cohort

This case–control study included patients between 18–49 years of age and undergoing surgeries for benign gynecological diseases between 1 June 2018 and 30 June 2019 in Linkou Chang Gung Memorial Hospital. Exclusion criteria included women who were postmenopausal, strict vegetarians, glucose intolerant, being medically treated for endocrine disorders, or had past or current diagnoses for gastrointestinal disorders or malignancies. Other baseline characteristics, such as parity, BMI, dietary preferences, prior medical (including hormone replacements) or surgical interventions, and current indications for surgeries were not subjected to exclusion. Written informed consent was obtained from all participants, and those with pathologically proven endometriosis formed the study group while the rest were placed in the control group. The study was reviewed and approved by the institutional review board of the Human Investigation and Ethical Committee of Chang Gung Medical Foundation (CGMHIRB No. 201601942B0).

### 4.2. Sample Collection

Recruited participants provided their urine and stool samples anytime during their hospitalization. Approximately 20–50 mL of urine was collected in a sterile, preservative-free container and kept at 4 °C. After dividing the specimen into two centrifugation tubes, each with 15 mL of urine, they were spun at 3000 rpm for 10 min. The upper liquid layer was separated into five 1.5 mL aliquots and placed in a 2 mL Eppendorf containing 2 mL NaN_3_. All specimens were kept at −80 °C until analyses for estrogen and metabolites.

Fresh fecal samples were collected in 150 mL, sterile Falcon tubes. After separating the stool into ten aliquots of two 20 mg, two 100 mg, and two 200–250 mg, five of the aliquots were bathed in RNAlater (QIAGEN Inc., Valencia, CA, USA) while the rest were placed in sterile phosphate buffer saline (PBS). All specimens were chilled at 4 °C and then frozen in liquid nitrogen within 3 h of retrieval. The fecal aliquots were also stored at −80 °C until DNA and protein extractions. 

### 4.3. Protein Extraction and Enzyme Activity Assay

Protein extraction was performed according to a published protocol [42]. Approximately 0.5 g of thawed feces was transferred into 10 mL conical tubes containing 5 mL of extraction buffer (60 mM Na_2_HPO_4_, 40 mM NaH_2_PO_4_, 10 mM KCl, 1 mM MgSO_4_) and vortexed for 1 min for homogenization. Sonication (Misonix Microson XL2000 Ultrasonic Homogenizer, Fisher Scientific, Pittsburgh, PA, USA) at 30 s intervals at maximum power was performed for lysis of bacterial cells. All the steps above were carried out in an ice bath. Then, supernatant containing extracted proteins was obtained after centrifugation of the lysates (15K rpm for 30 min) at 4 °C. Activities of β-glucuronidase (Fluorometric, Abcam, Cambridge, UK) and β-glucosidase (Sigma-Aldrich, Darmstadt, Germany) were measured using commercial activity assay kits and following the manufacturers’ protocols. Briefly, a 96-microplate (Thermo Scientific, Waltham, MA, USA) placed with 5–20 μL fecal lysate and adjusted to 90μL with assay buffer for each reaction was prepared. After addition of the provided Substrate Mix (Abcam, Cambridge, UK) for 0–60 min at 37 °C, fluorescence was measured (Ex/Em = 330/450 nm) with multi-well enzyme-linked immunosorbent assay (ELISA) spectrophotometer (Tecan Infinite M200 Plate Reader, Ramsey, MN, USA). 

### 4.4. Genomic DNA Extraction and 16S rRNA Gene Sequencing

Total genomic DNA from fecal samples was extracted using the column-based method (QIAamp PowerFecal DNA Kit, Qiagen, Valencia, CA, USA) based on manufacturer’s instructions, and DNA concentration was determined and adjusted to 5 ng/ul for polymerase chain reaction (PCR) amplification and purification. In accordance with the 16S Metagenomic Sequencing Library Preparation procedure (Illumina), V3-V4 region of the 16S rRNA gene was amplified with specific primer set (341F: 5′-CCTACGGGNGGCWGCAG-3′, 806R: 5′- GACTACHVGGGTAT CTAATCC -3′) [50]. In brief, 12.5 ng of gDNA was used for the PCR reaction (KAPA HiFi HotStart ReadyMix, Roche, Indianapolis, IN, USA) under the following condition: 95 °C for 3 min; 25 cycles of 95 °C for 30 s, 55 °C for 30 s, and 72 °C for 30 s; and 72 °C for 5 min and hold at 4 °C. Samples with a bright main strip (around 500 bp) in 1.5% agarose gel were chosen and purified with AMPure XP beads for library preparation and sequencing (Illumina MiSeq platform).

Sequences (300 bp) from amplicon sequencing using FLASH v1.2.11 [51] were chimera-checked using UCHIME (http://www.drive5.com/usearch/manual/uchime_algo.html, accessed on 1 June 2021) to obtain effective tags [52] and filtered from the data set before operational taxonomic unit (OTU) clustering at 97% sequence identity using the UPARSE [53] function in the USEARCH pipeline v7.0.1090 [54].

### 4.5. Liquid Chromatography and Tandem Mass Spectrometry

For analyses of urine and fecal estrogens and metabolites, samples were transferred on dry ice to Biotools Laboratory located in New Taipei City, Taiwan. The lab used Agilent 1290 ultra-high-performance liquid chromatography system (Agilent Technologies, Santa Clara, CA, USA) for chromatographic separation of target compounds. Agilent Eclipse Plus C18 column (2.1 × 100 mm, 1.8 μm), set at 40 °C, was used for separation of each sample injection of 20 μL. Then, mass spectrometry analysis was completed with Agilent 6470 triple quadrupole mass spectrometer equipped with an AJS-ESI ion source in multiple reaction monitoring (MRM) mode. Based on their signal-to-noise ratios, the detection and quantitation limit of the method were calculated. Lower limit of detection (LLOD) was defined as the compound concentration corresponding to a signal-to-noise ratio of 3 while the lower limit of quantitation (LLOQ) was defined as the compound concentration corresponding to a signal-to-noise ratio of 10, following the US FDA guideline for bioanalytical method validation. Final concentrations, presented in ng/mL, were measured for 14 target metabolites: 2-hydroxyestrone, 2-methoxyestrone, 4-hydroxyestrone, 16-epiestriol, 4-methoxyestradiol, 2-hydroxyestradiol, 16α-hydroxyestrone, 2-methoxyestradiol, 4-methoxyestrone, 17-epiestriol, 2-hydroxyestrone-3-methyl ether, β-estradiol, estriol, and estrone.

### 4.6. Statistical Analysis

Using the QIIME pipeline v1.9.1, subsequent analyses of alpha and beta diversities were both performed using the normalized data. Alpha diversity was indicative of the species complexity within individual samples, represented with Shannon, Simpson, Chao1, and good coverage. Differences among samples in terms of species complexity were demonstrated with two beta diversity parameters, the weighted and unweighted UniFrac [55]. Boxplots were created using free statistical package R3.1.0 while Principal Coordinates Analysis (PCoA) analysis was conducted with the WGCNA, stat, and ggplot2 packages in R3.1.0. For functional analysis, functional abundances from 16S rRNA sequencing data were analyzed for the prediction of functional genes with Phylogenetic Investigation of Communities by Reconstruction of Unobserved States (PICRUSt, v1.1.1). For statistical analysis, significance of all species among groups at various taxonomic level were detected using differential abundance analysis with a zero-inflated Gaussian (ZIG) log-normal model as implemented in the “fitFeatureModel” function of the Bioconductor metagenomeSeq package [56]. In addition, linear discriminant analysis effect size (LEfSe), which utilizes a non-parametric Kruskal–Wallis test and Wilcoxon rank-sum test to identify bacterial taxa whose relative abundance is significantly different between endometriosis and control, was indicated with linear discrimination analysis (LDA). In this study, taxa with LDA score (log 10 > 3) were considered significant, and Analysis of similarities (ANOSIM) and MRPP were used to determine whether the community structures significantly differ among and within groups. For differences in each taxon level (phylum, class, order, family genus), Welch’s *t*-test was performed using the Statistical Analysis of Metagenomic Profiles (STAMP) software v2.1.3. A *p*-value < 0.05 was considered significant for all statistical tests employed.

## 5. Conclusions

By comparing the stool and urine samples of a specific cohort of Taiwanese women with or without histologically proven endometriosis, our results add to the body of literature regarding the microbiota of people with this chronic condition. In addition to incorporating enzymatic assays and estrogen metabolite quantifications to the microbial studies, this study has the advantage of evaluating a population of women with similar background (same ethnicity, residence, age, and dietary sources). Although no significant differences were identified in fecal β-glucuronidase activities, alpha diversity, beta diversity, indicators of dysbiosis, and overall microbial compositions in patients with or without the disease, the relative abundance of *Erysipelotrichia* class, along with fecal folds of four estrogens/estrogen metabolites, were significantly higher in the endometriosis group. While these findings are not enough to establish potential biomarkers or specific targets for therapeutic interventions, such as probiotics or antibiotics, more extensive research is evidently needed to characterize the gut microbiota and clarify its involvement in endometriosis.

## Figures and Tables

**Figure 1 ijms-24-16301-f001:**
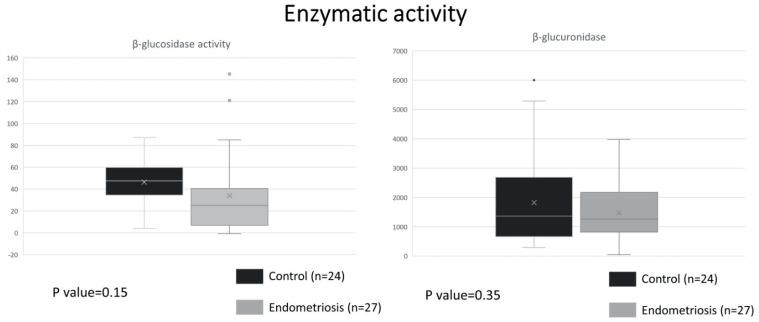
Enzymatic assay with results given in U/L. Boxplots: box indicate the 1st and 3rd quartiles, dash lines the upper and lower whiskers, cross indicates mean, horizontal bold lines the median, and dots signifies outliers.

**Figure 2 ijms-24-16301-f002:**
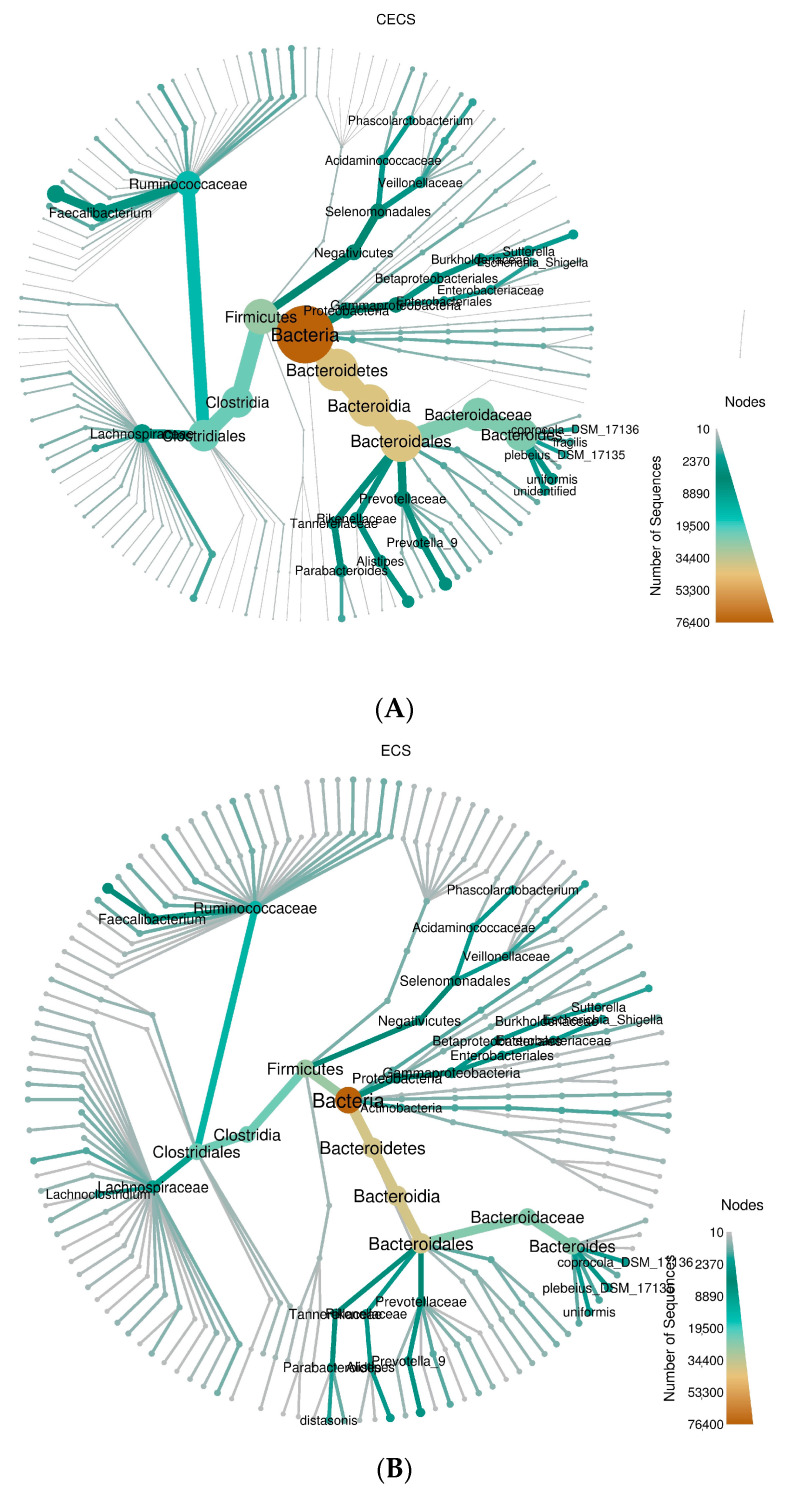
A circular heat tree represents the sequence abundance of different hierarchical taxa, displayed from the center outwards. The innermost circle represents the highest taxonomic level, the bacterial domain (Bacteria). Moving outward from the center, the taxonomic levels decrease, and the number of sequences annotated to different taxonomic levels decreases accordingly. Sequence abundance is represented by node size, branch thickness, and color. Species with higher abundance are indicated by larger nodes, thicker branches, and colors closer to brown. The legend in the lower-right corner shows the sequence count along with their corresponding colors and node sizes. (**A**) Control and (**B**) Endometriosis. CECS = control, ECS = endometriosis.

**Figure 3 ijms-24-16301-f003:**
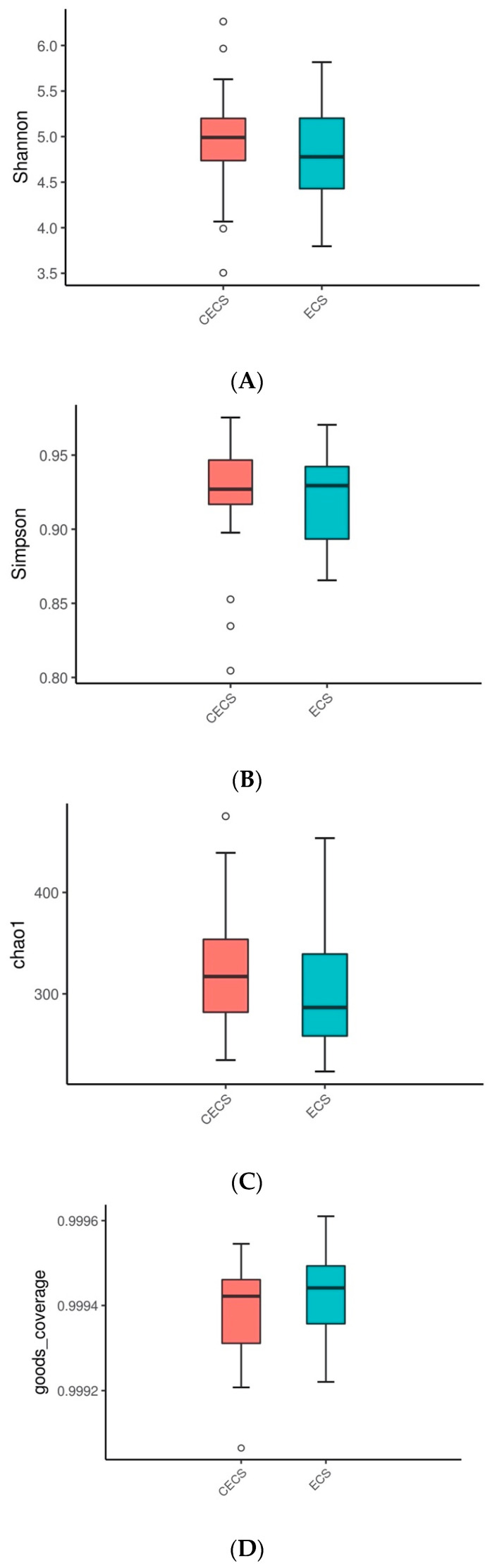
Alpha diversity analysis on a single sample reflects the richness, evenness, and diversity of microbial communities within that sample. Shown in box plots, no significant differences were observed using Shannon (**A**) and Simpson (**B**) indices, which indicated that the alpha diversity of gut microbiota was similar in the endometriosis and control group. Chao analysis (**C**) detected similar microbial richness between the two groups. The >99% Good’s coverage index (**D**) demonstrated that most of the gut microbial taxa were identified. *T*-test was used to calculate for significant differences. CECS = control, ECS = endometriosis.

**Figure 4 ijms-24-16301-f004:**
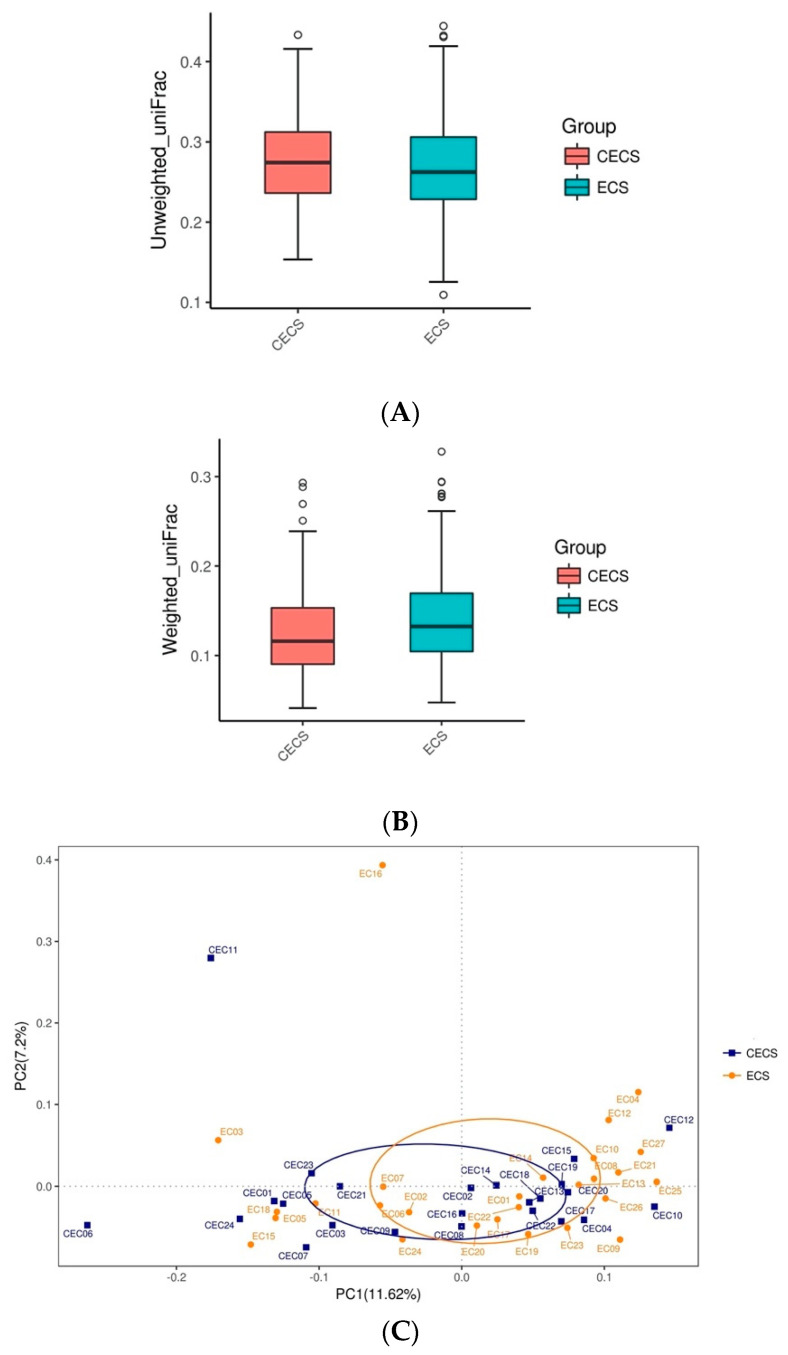
Beta diversity studies use OTU representative sequences to construct phylogenetic trees and calculate unweighted UniFrac and weighted UniFrac (when relative abundance of species within sample was considered) distances in order to evaluate differences between samples. Smaller values indicate less differences in species diversity between samples. Results are represented with box plots: (**A**) unweighted UniFrac and (**B**) weighted UniFrac. *T*-test was used to calculate for significant differences. Principal coordinate analysis plots (PCoA) at the OTU level based on the Bray-Curtis dissimilarity matrix demonstrated that for both unweighted (**C**) and weighted UniFrac distances (**D**), gut microbiota did not appear to cluster by presence or absence of the disease (control vs. endometriosis). Samples with similar species composition would cluster together while those with less similarities would be further apart in distance. CECS = control, ECS = endometriosis.

**Figure 5 ijms-24-16301-f005:**
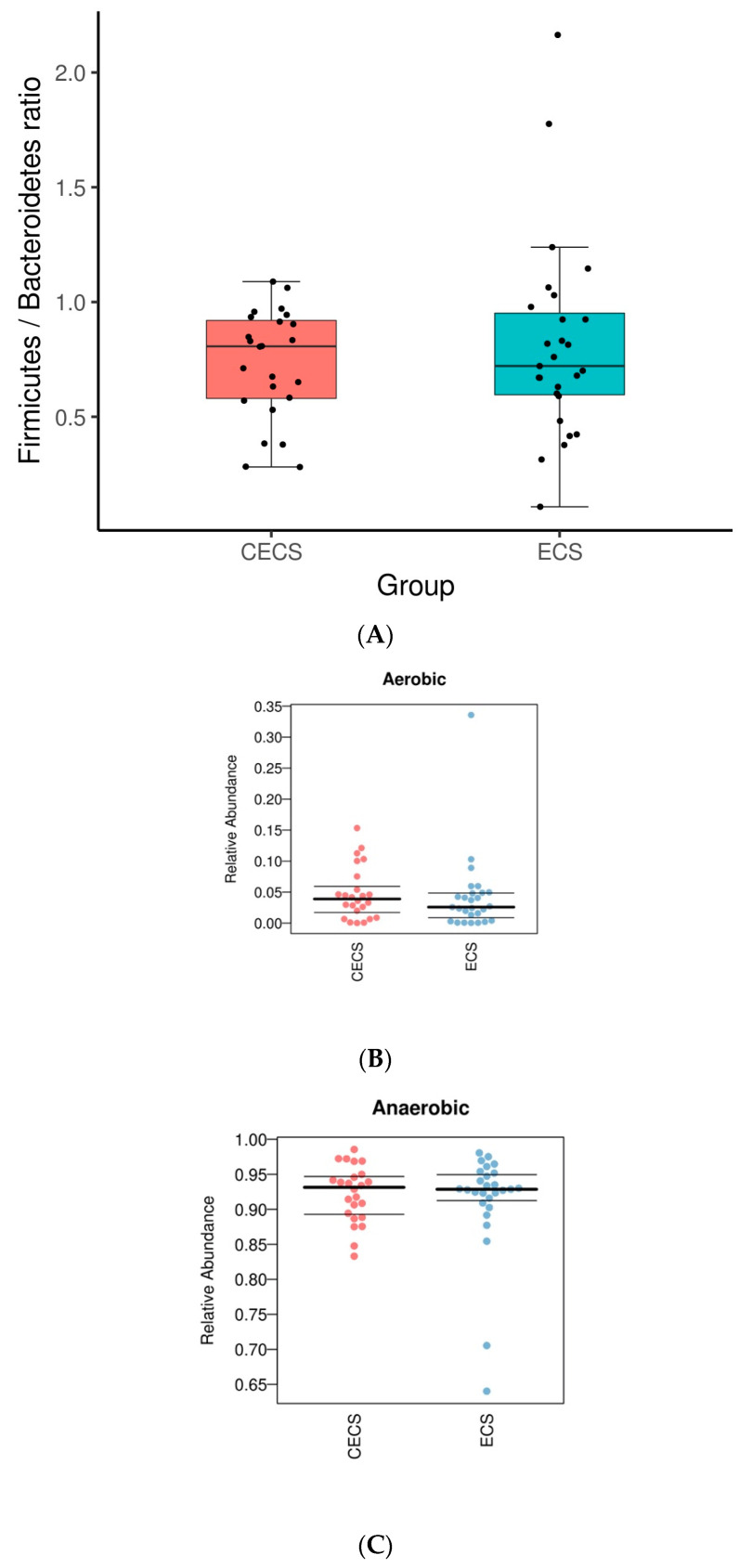
(**A**) F/B (Firmicutes/Bacteroidetes) ratio of the control and endometriosis group shown in box plot. BugBase application uses sequencing data for microbial phenotype prediction, which suggested a similar abundance of aerobic (**B**) and anaerobic (**C**) bacteria in both groups of patients. *T*-test was used to calculate for significant differences. (**D**) Linear discrimination analysis (LDA) coupled with effective size measurements, using 3.0 as threshold for discriminative features and *p* < 0.05 for statistical tests, identified the most differentially abundant taxa between the two groups. Comparison of relative abundance at the bacteria (**E**) class, (**F**) order, (**G**) family, and (**H**) genus levels between the two groups are shown. CECS = control, ECS = endometriosis.

**Table 1 ijms-24-16301-t001:** Baseline characteristics of control and endometriosis group.

Characteristics	Control (*n* = 24)	Endometriosis (*n* = 27)	*p*-Value
Age, yr (mean ± SEM)	37.7 ± 1.3	38.1 ± 1.0	0.770
BMI, kg/m^2^ (mean ± SEM)	24.04 ± 0.87	21.77 ± 0.73	0.051
Gravida (median, IQR)	1.0 (0.0–2.0)	1.0 (0.0–2.0)	0.825
Parity (median, IQR)	0.0 (0.0–1.75)	0.0 (0.0–1.0)	0.975
CA-125, U/mL (mean ± SEM)	21.54 ± 3.12	70.07 ± 9.07	<0.001 *
rAFS score (mean ± SEM)	NA	31.1 ± 6.4	NA
Endometriosis stage (%) I	NA	29.6 (8/27)	NA
II	NA	14.8 (4/27)	NA
III	NA	25.9 (7/27)	NA
IV	NA	29.6 (8/27)	NA

NA: not applicable. * *p*-value (s) with statistical significance.

**Table 2 ijms-24-16301-t002:** Elevations of estrogens and estrogen metabolites in fecal and urinary samples of patients with or without endometriosis.

	Stool		Urine	
	Fold (Endometriosis/Control)	*p*-Value	Fold (Endometriosis/Control)	*p*-Value
Estradiol	2.093	0.121	1.218	0.224
Estriol	2.002	0.011 *	N.D.	N.D.
Estrone	1.220	0.059	0.756	0.902
16-epiestriol	1.681	0.018 *	1.292	0.183
16α-hydroxyestrone	2.991	0.016 *	2.894	0.404
17-epiestriol	0.834	0.21	1.189	0.328
2-hydroxyestradiol	0.851	0.829	0.968	0.546
2-hydroxyestrone	1.049	0.282	0.890	0.794
4-hydroxyestrone	0.669	0.468	0.941	0.962
2-methoxyestradiol	1.799	0.035 *	N.D.	N.D.
4-methoxyestradiol	1.578	0.196	N.D.	N.D.
2-methoxyestrone	0.389	0.197	1.566	0.374
4-methoxyestrone	N.D.	N.D.	1.712	0.595
2-hydroxyestrone-3 methyl ether	0.558	0.84	N.D.	N.D.

N.D. = not detected. * *p*-value (s) with statistical significance.

## Data Availability

The data presented in this study are available on request from the corresponding author. The data are not publicly available due to containment of patient identifiers.

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
