# Peer review of "Gut Microbiome–Estrobolome Profile in Reproductive-Age Women with Endometriosis"

_ijms, 2023, doi:10.3390/ijms242216301_

Round 1

Reviewer 1 Report

Comments and Suggestions for Authors

Some specific comments are as follows:

Suggest adding a paragraph in the Introduction section about endometriosis’s stages.

Line 77: define the abbreviation that is mentioned for the first time (BMI), and any other abbreviation within the whole text.

Primers used in PCR need reference or it is designed by the author.

Suggest adding the “Supplemental Figure 1” as a main figure within the article and not supp.

Figures 1 (A-G) are too many and give the same idea, choose the most significant ones within the text and the rest as supplementary.

Reviewer 2 Report

Comments and Suggestions for Authors

Comments to the authors

While the link between gut microbiota and endometriosis has been shown in very recent studies, there is a lack of sufficient quantity and quality of information within the existing literature on this topic. Overall, this is a well-written manuscript; however, there are some suggestions that authors should consider before making progress:

The global prevalence rate of endometriosis at reproductive age should be added to the introduction.

In lines 46–47, the authors should mention other factors that are involved in the pathogenesis of endometriosis, such as immunological, genetic, hormonal, lifestyle-related, and environmental factors.

In line 72, the authors should specify the age range of patients.

Although there is a link between gut microbiota and obesity, body mass index was not considered an exclusion criteria. The authors should explain the reason(s) or include that in the limitations of their study.

There are similar articles on this topic that are not cited in this manuscript; I would suggest citing those articles.

A great systematic review by Talwar et al. (2022): “The gut microbiota: a double-edged sword in endometriosis”.

An original article by Ni et al. (2020): “Correlation of fecal metabolomics and gut microbiota in mice with endometriosis”.

Finally, the authors should clarify whether there is a potential gut microbiome-based therapy for endometriosis based on their findings.
